# Prevalence of *Dichelobacter nodosus* and Ovine Footrot in German Sheep Flocks

**DOI:** 10.3390/ani11041102

**Published:** 2021-04-12

**Authors:** Julia Storms, Anna Wirth, Danae Vasiliadis, Isabelle Brodard, Antje Hamann-Thölken, Christina Ambros, Udo Moog, Jörg Jores, Peter Kuhnert, Ottmar Distl

**Affiliations:** 1Institute of Animal Breeding and Genetics, University of Veterinary Medicine Hannover (Foundation), 30559 Hannover, Germany; Julia.storms@tiho-hannover.de (J.S.); Anna.Maria.Wirth@tiho-hannover.de (A.W.); Danae.vasiliadis@tele2.de (D.V.); 2Institute of Veterinary Bacteriology, Vetsuisse Faculty, University of Bern, 3012 Bern, Switzerland; isabelle.brodard@vetsuisse.unibe.ch (I.B.); joerg.jores@vetsuisse.unibe.ch (J.J.); peter.kuhnert@vetsuisse.unibe.ch (P.K.); 3Animal Health Services of Lower Saxony, 26121 Oldenburg, Germany; antje.hamann-thoelken@lwk-niedersachsen.de; 4Bavarian Animal Health Services, 91522 Ansbach, Germany; christina.ambros@tgd-bayern.de; 5Animal Disease Fund Thuringia, 07745 Jena, Germany; umoog@thueringertierseuchenkasse.de

**Keywords:** footrot, sheep, prevalence, *Dichelobacter nodosus*, Germany, real-time PCR

## Abstract

**Simple Summary:**

Footrot is a highly contagious foot disease in sheep and a common cause of lameness. It is a major challenge for sheep industries worldwide and has great economic impact on production. Due to the pain associated with the disease, it is considered an animal welfare issue. Footrot is caused by the bacterium *Dichelobacter nodosus* (*D. nodosus*), which encompasses benign and virulent strains. Benign *D. nodosus* commonly causes an inflammation of the interdigital skin whereas virulent strains can lead to severe footrot with a separation of hoof horn from the underlying soft tissue as the disease progresses. The objectives of this field study were to determine the prevalence of *D. nodosus* in a wide range of sheep flocks across Germany using swab samples from the interdigital skin of the feet. Due to the high prevalence of 42.93% of *D. nodosus* in the German sheep population, further work is required to determine measures on how to decrease the prevalence.

**Abstract:**

The bacterium *Dichelobacter nodosus* (*D. nodosus*) is the causative agent of ovine footrot. The aim of this field study was to determine the prevalence of *D. nodosus* in German sheep flocks. The sheep owners participated voluntarily in the study. More than 9000 sheep from 207 flocks were screened for footrot scores using a Footrot Scoring System from 0 to 5 and sampling each sheep using one interdigital swab for all four feet of the sheep. The detection and discrimination between benign and virulent strains was done employing a real-time PCR. Our results showed a mean prevalence of 42.93% of *D. nodosus* in German sheep on an animal level. Underrunning of hoof horn on at least one foot (Scores 3-5) was detected in 567 sheep (6.13%). Sheep with four clinically healthy feet were found through visual inspection in 47.85% of all animals included in this study. In total, 1117 swabs from sheep with four clinically healthy feet tested positive for *D. nodosus*. In 90.35% of the positive swabs, virulent *D. nodosus* were detected. Benign *D. nodosus* were detected in 4.74% of the *D. nodosus*-positive swabs while 4.91% tested positive for both, benign and virulent *D. nodosus*. In 59 flocks *D. nodosus* were not detected and in 115 flocks only virulent *D. nodosus* were found while seven flocks tested positive for benign strains.

## 1. Introduction

Ovine footrot is a highly contagious disease that affects the feet of sheep and other ruminants and it is a major challenge for sheep industries worldwide [1]. The aetiological agent is the gram-negative, anaerobic but aerotolerant bacterium *Dichelobacter nodosus* (*D. nodosus*) [2]. 

The disease has great economic impact on production due to decreased wool quality, reduced live-weight gain and decreased lambing percentages [3]. Ovine footrot is a painful disease, and thus, considered to constitute a significant animal welfare issue [4]. Therefore, eradication of *D. nodosus* from the flock is beneficial and feasible, provided sufficient commitment, availability of labor and financial means by the sheep owner [5]. Additional factors like the local climate conditions, production systems and biosecurity status of the flock also play a role in the elimination of *D. nodosus* [6].

The clinical signs can range from mild interdigital dermatitis to the complete underrunning of the hoof horn and the loss of the hoof capsule in severe cases [7]. Lameness is often observable in affected animals, but it is not a consistent sign [7]. A grey pasty scum in the interdigital space and a foul smell are characteristic clinical signs [8]. Major differential diagnoses are contagious ovine digital dermatitis, white line disease and foot abscesses [9]. There is a seasonal impact on the disease’s progression. Warm and moist environmental conditions favor the transmission of *D. nodosus* between sheep [10] and the development of clinical signs in the presence of *D. nodosus* [11]. 

Benign and virulent strains of *D. nodosus* exist and can be differentiated based on the respective extracellular proteases AprV2 and AprB2 they produce. There is a 2-bp substitution in the corresponding genes *aprV2* and *aprB2*, which results in a single amino acid change [12]. The competitive real-time PCR method developed by Stäuble et al. [13] uses this variation in the *aprV2* and *aprB2* genes for the simultaneous detection and discrimination of virulent and benign *D. nodosus*. 

The objectives of this nationwide study were to determine the prevalence of footrot and its causative agent *D. nodosus* in German sheep flocks with the use of real-time PCR. In addition to the presence or absence of *D. nodosus* the clinical diagnosis depends on environmental factors and host susceptibility [14]. Therefore, laboratory diagnosis with the use of the highly sensitive and specific real-time PCR method for determining virulent and benign *D. nodosus* is ideally involved [14]. A broad range of different farms was included in this field study in order to reflect the diverse picture of sheep farms in Germany. 

## 2. Materials and Methods

### 2.1. Ethical Approval

The study was approved by the Institutional Animal Care and Use Committee (IACUC) (33.19-42502-05-19A414) and the respective state veterinary offices from the different German states. The sampling and handling of the sheep followed European Union guidelines for animal care and handling and the Guidelines of Good Veterinary Practices.

### 2.2. Sample Collection 

We announced the project broadly in several meetings of sheep breeding organizations all over Germany and sheep breeders’ journals (Schafzucht, Schäferbrief) and published the project on our website (https://www.tiho-hannover.de/kliniken-institute/institute/institut-fuer-tierzucht-und-vererbungsforschung/forschung/forschungsprojekte-schaf/moderhinke-mores (accessed on 10 January 2019)). All sheep owners were invited to participate in the study. The sheep owners, who were interested in participating, gave written consent. Furthermore, information was collected concerning the flock size and the footrot history. The participating farms were assigned to one of three study areas according to their location. The three study areas were defined as North Germany (Schleswig-Holstein, Lower Saxony, North Rhine-Westphalia), East Germany (Brandenburg, Saxony, Saxony-Anhalt, Thuringia) and South Germany (Bavaria, Baden-Württemberg, Hesse, Rhineland-Palatinate) (Table 1, Figure 1). 

In total, samples of 9243 sheep from 207 flocks were included in the study. The flock sizes ranged from 10–2400. The flock size was defined as the number of ewes and rams at the time of sampling. The samples were collected from January 2019 until September 2020. The mean sample size per flock was 44.7 and sampling of animals within a flock was randomized (Table 2). Mostly ewes and yearlings and all rams of the flock were sampled. All sheep were sampled in 60 flocks (whole-flock sampling). For classifying clinical lesions, the Footrot Scoring System of the Swiss Consulting and Health Service for Small Ruminants was applied (Table 3) [15].

The sample collection was performed by veterinarians. The sheep were turned over manually or via a sheep chair, tilt table or a handling system. For the mobile data collection the APR600 ISO11784/11785 RFID Handheld Reader (Agrident, Barsinghausen, Germany), which is compatible for both transponder types HDX and FDX-B, was used. An individual Task mode was preprogramed to fit our system. The electronic ear tag of every sheep was scanned and further information was recorded. Each swab sample was labelled with a unique identification code. The gross dirt was removed from the interdigital space and every foot was inspected. A sterile and dry cotton swab (101 × 16.5 mm) (Sarstedt, Nümbrecht, Germany) in a tube with a screw top and without a transport medium was used to sample the interdigital skin of the foot. The cotton swab was turned 90° and used for the second interdigital space. This procedure was repeated until all four feet were sampled using one swab. The pooling of four individual foot samples of the same animal into one 4-feet sample is an adequate method to reduce the real-time PCRs of individual sheep [16,17]. The screw top of the tube with the cotton swab within was closed and stored safely in a carton box. The highest footrot score diagnosed was recorded and saved in the electronic reader.

### 2.3. Laboratory Analysis

The samples were stored at −20 °C until processing. The first step of the process was the thawing of the cotton swabs at room temperature for one hour. DNA was isolated with the Qiagen DNeasy Blood and Tissue Kit (Qiagen, Hilden, Germany). For the detection and discrimination of benign and virulent *D. nodosus* the real-time PCR developed by Stäuble et al. [13] was performed. Control strains for virulent and benign *D. nodosus* were the type strain ATCC 25549T and the field isolate JF5922, respectively. The benign control strain was isolated at the Institute of Veterinary Bacteriology, University of Bern, Switzerland [18]. The 25-µL reaction mixture containing 22.5 µL TaqMan Fast Advance MasterMix (Life Technologies GmbH, Thermo Fisher Scientific, Waltham, Massachusetts) and 2.5 µL sample DNA (10–100ng) were pipetted in duplicates into a 96-well plate. Two nontemplate samples (pyrogen-free water) as negative controls were studied in each qPCR run. The amplification was carried out in a 7500 Real-Time PCR-System (ABI, Thermo Fisher Scientific, Waltham, MA, USA) and applying the cycle instructions according to Stäuble et al. [13].

The results were analyzed using the QuantStudio 3 System Software (Thermo Fisher Scientific, Waltham, MA, USA) with the threshold set at 0.06500. Samples with a Ct-value < 40 were defined as being positive and a Ct ≥ 40 was interpreted as negative. 

## 3. Results

### 3.1. Distribution of Footrot Scores

In total, 9243 interdigital swab samples were collected from 207 sheep farms. The examination of feet showed healthy interdigital skin and claws (Score 0) in 4423 sheep (47.85%). Interdigital dermatitis (Score 1 and 2) was found in 4253 sheep (46.02%). There were 567 sheep (6.13%) which showed a separation of hoof horn from the underlying dermis (Score 3–5) (Table 4).

### 3.2. Real-Time PCR Results 

The real-time PCR method detected *D. nodosus* on 3968 sheep (42.93 %). Sheep with only virulent (*aprV2+*) strains of *D. nodosus* were more frequent (90.35%) than sheep with only benign (*aprB2+*) strains (4.74%) while 4.91 % of sheep harbored both (*aprV2+/aprB2+*) virulotypes (Table 5, Appendix A). No *D. nodosus* were detected in 5275 swab samples (57.07%) (Table 5).

In 59 of the 207 sheep flocks, no *D. nodosus* were detected and in 148 sheep flocks *D. nodosus* were present in a variable flock prevalence ranging from 0.5–100% (Appendix A). 

In the swab samples from healthy feet (Score 0), 3306 (74.75%) were negative for *D. nodosus*. Virulent *D. nodosus* were detected in 22.59% of sheep with Score 0. For Score 1, the results showed 1878 samples (52.93%) with no *D. nodosus* and 1466 samples (41.32%) with virulent *D. nodosus*. Virulent *D. nodosus* were detected in more than 97% of feet showing underrunning of the hoof horn (representing Score ≥ 3), while benign *D. nodosus* (*aprB2+*) were not detected in these samples alone. Benign *D. nodosus* could be found neither alone (*aprB2+*) nor in combination with virulent *D. nodosus* (*aprB2+/aprV2+*) in samples of scores 4 or 5 (Table 6).

In 165 of the 207 flocks (79.71%) all sheep had clinically healthy feet (Score 0) or showed only low footrot scores of 1–2. Higher footrot scores of 3–5 with an underrunning of hoof horn were present on the feet of sheep in 42 flocks (20.21%). In the 59 flocks (28.50%), in which *D. nodosus* were not detected, no sheep with a separation of hoof horn (Footrot scores 3–5) were found. In 17 flocks (8.21%), only sheep with clinically healthy feet (Score 0) were found, but swab samples tested positive for *D. nodosus*. No feet with underrunning of the hoof horn (Score 3–5) were identified in any of the seven flocks (3.38%), in which only benign *D. nodosus* (*aprB2+*) were detected (Table 7).

## 4. Discussion

The prevalence of footrot and its causative agent *D. nodosus* has been the subject of different studies in the past. In Germany, to the best of our knowledge, studies to determine the prevalence of *D. nodosus* have not been conducted yet. 

The current study was announced broadly across Germany and it was stated that every sheep farmer was able to participate. Sheep owners with and without an apparent footrot problem participated voluntarily. Out of >300 submissions from different sheep owners, 207 sheep flocks were visited, and more than 9000 swab samples were collected. Therefore, this is the largest study so far investigating the prevalence of ovine footrot. Due to the voluntary participation the sample collection was not completely randomized, which has to be considered in the interpretation of the results. Although it was emphasized that all sheep farmers irrespective of their footrot status could participate in the study, the possibility cannot be ruled out that there might be greater motivation of those sheep farmers whose sheep were affected by footrot. Also, the distribution of sheep flocks was not equal across the states of Germany, due to the logistics of the sample collections. Nevertheless, the combination of 9243 analyzed swab samples and the large number of flocks suggests that the results of the study are likely to be robust.

The real-time PCR method we used to determine the presence of *D. nodosus* on the feet of the sheep was also successfully used in other studies [16,17,19,20,21]. It has proved to be sensitive and specific with a sensitivity of 100% to detect *D. nodosus* [13].

In our study, 3548 sheep showed a mild inflammation of the interdigital skin (Score 1) and 1878 (52.93%) of these sheep tested negative for *D. nodosus* (Table 6). The early stages of footrot cannot be distinguished from interdigital dermatitis of another cause by clinical signs alone [22]. Therefore, in these cases the diagnosis of footrot can only be made in retrospect after the detection of *D. nodosus*. On the other hand 1117 sheep with clinically healthy feet (Score 0) tested positive for *D. nodosus*. This can be explained, because footrot is a multifactorial disease. The virulence of *D. nodosus* has an impact on the capacity of causing severe lesions, while recent studies suggest that the presence of the *aprV2* gene in *D. nodosus* isolates may not be the only characteristic determining the clinical outcome of lesions [23]. In addition, environmental conditions and the inherent susceptibility of the sheep play a role in the expression of clinical footrot [7]. In 17 sheep flocks (8.21%) only clinically healthy sheep were present, but swab samples tested positive for *D. nodosus*. An explanation may be that environmental conditions or other factors were unfavorable for the bacterium. This finding emphasizes the contribution of laboratory diagnosis using qPCR in determining the presence or absence of *D. nodosus* and its prevalence at times without clinical signs of footrot on sheep. Sheep farmers could apply this knowledge in order to implement certain management practices, like a close monitoring of sheep or other preventative actions to avoid future footrot outbreaks. Since no sheep flocks were revisited and no sheep were reinspected after the sampling, the development of the clinical presentation of feet could not be recorded. Mild foot lesions or clinically healthy feet at the time of sampling may have been the early stages of footrot and could have progressed to higher footrot scores later on [24].

Similar to our results, a questionnaire done in 2012 by Friedrich et al. [25] also showed that footrot is common in many sheep flocks in Germany, Austria and Switzerland. The disease is of significance today in these countries. A within-flock prevalence of 36 ± 31% was determined, whereby the flock prevalence of *D. nodosus*, which was determined in our study, showed a greater variability, ranging from 0.5–100% (Appendix A). Footrot was observed in 66% of the German sheep flocks, which participated in the questionnaire, while we found that only 42 of the 207 flocks (20.29%) kept sheep with footrot scores of 3–5 at the time of sampling, which can be interpreted as being equivalent to clinical footrot. The results of the questionnaire may be an underestimation because the sheep farmers diagnosed the disease themselves in 89%, which is not as specific as the implementation of real-time PCR as a diagnostic tool to determine the presence of *D. nodosus* in early stages and mild cases of footrot, thus securing the diagnosis. Furthermore, the climate, host susceptibility and management practices affect the clinical appearance of feet in the presence of *D. nodosus*. Hence, the different factors, which influence the severity of lesions, need to be considered comparing the prevalence of footrot scores. 

Compared to other countries, which studied the frequency or distribution of *D. nodosus,* the prevalence of 42.93% that was determined in this field study is in the upper range. Results of previous studies on the prevalence of *D. nodosus* in other countries cannot be easily compared, due to the differences in methods and data collection (Appendix A).

In Switzerland the prevalence of virulent *D. nodosus* in sheep was estimated 16.9% and for benign *D. nodosus* the prevalence was 6.3% [17]. The greater prevalence of virulent strains compared to benign strains is similar to our findings, although in our study the prevalence of benign *D. nodosus* was lower compared to the prevalence in Switzerland. The overall lower prevalence of *D. nodosus* in sheep in Switzerland may be due to the different method of complete randomization using a two-stage cluster sampling strategy and only sampling five animals per farm in the Swiss study. In addition, the sample size was smaller. Another reason could be that footrot control measures first started in 1990 and a nationwide footrot control program is currently being put into action [17].

A study that was conducted on Swedish slaughter lambs in abattoirs showed a prevalence of footrot of 5.8% and it was also found that 97% of footrot affected feet tested positive for *D. nodosus* by PCR [26]. This estimated prevalence is lower than the prevalence in Germany. However, it might be an underestimation as lambs with severe footrot are not sent to abattoirs and only 60 samples (feet with scores ≥ 2) were examined by PCR. Thus, the actual prevalence of *D. nodosus* might be higher because our results show high frequencies of virulent *D. nodosus* on healthy feet (Score 0) and feet with mild interdigital inflammation (Score 1). Also, the methodology of sampling only slaughter lambs in abattoirs and no ewes may not always represent a section of the flock.

In accordance to our results, another Swedish study showed that *D. nodosus* was more commonly found in feet with footrot than in healthy feet. However, in contrast to our findings, benign strains are more frequent than virulent strains in Sweden [27].

Consistent with our results, Maboni et al. [28] detected *D. nodosus* in 46 biopsy samples from 79 healthy feet from UK sheep (58%). Further, virulent strains were more frequent than benign strains (7%) [28]. Winter et al. [29] used a different approach, when they conducted a survey on the prevalence of lameness in England. In this study, clinical footrot was classified into interdigital dermatitis (ID) and severe footrot (SFR). It was concluded that 90% of lame sheep showed either ID or SFR and 80% of farmers reported that footrot is the most common cause of lameness. ID has a mean prevalence of 4.5% and SFR of 3.1% in England. Applying this classification to our footrot scoring system, the clinical signs of ID are equivalent to the definition of our footrot scores 1–2 and severe footrot therefore equals our footrot score 3–5. Compared to this, we detected a higher prevalence of ID by factor 10 (45.14%) and SFR by factor 2 (6.22%). However, in this study in 59 sheep flocks (28.5%) *D. nodosus* were not detected and in 165 flocks (79.71%) no sheep with high footrot scores of 3–5 were present at the time of sampling, whereas in England almost all flocks were affected by footrot. Comparing these two studies, the different methods (questionnaire versus qPCR) need to be considered. The prevalence of ID might be underestimated by sheep farmers in England as a mild dermatitis may be overlooked if affected sheep don’t show signs of lameness.

The prevalence of footrot in Norway is not comparable to other countries, where footrot is endemic in the sheep population because it was newly reintroduced to the country in 2008 after 60 years of freedom from disease. The occurrence of footrot was confined to the county of Rogaland with a prevalence of 1.5% [30]. The surveillance program that started in 2014 aims at detecting ovine footrot in the field and at slaughterhouses. The occurrence of severe footrot has been low in previous years in Norway [31].

In 1988, the New South Wales (NSW) Strategic Footrot Plan was implemented in NSW, Australia with the aim to progressively eliminate virulent footrot. The prevalence of footrot dropped below 1% in all areas in 2009 [32]. The prevalence increased to >1% in some areas in 2018 [33]. This lower prevalence in NSW may be because of the positive impact of the eradication program. On the other hand, this prevalence may be underestimated, because the surveillance mostly relies on clinical inspections on saleyards or farms. 

## 5. Conclusions

The results of this study show that footrot and its causative agent, *Dichelobacter nodosus* are common in sheep flocks across Germany. The great voluntary participation in the study indicates that footrot is an important issue for sheep owners as well. This supports the fact that footrot has a negative impact on sheep production and animal welfare.

Our data confirmed that *D. nodosus* were much more common in sheep flocks than clinical signs of footrot and demonstrated that the PCR method in detecting *D. nodosus* is more sensitive than the visual inspection of feet alone. Therefore, a widespread implementation of real-time PCR as a diagnostic tool would be useful to determine the presence or the prevalence of *D. nodosus* in sheep flocks with uncertain clinical appearance on feet and low footrot scores. Thus, the potential risk for sheep to develop severe footrot can be assessed, appropriate measures can be taken and hence potential future outbreaks might be prevented.

Due to the high prevalence of *D. nodosus* in the German sheep population as shown in this study, further research, including effective ways to reduce the prevalence, is needed. 

## Figures and Tables

**Figure 1 animals-11-01102-f001:**
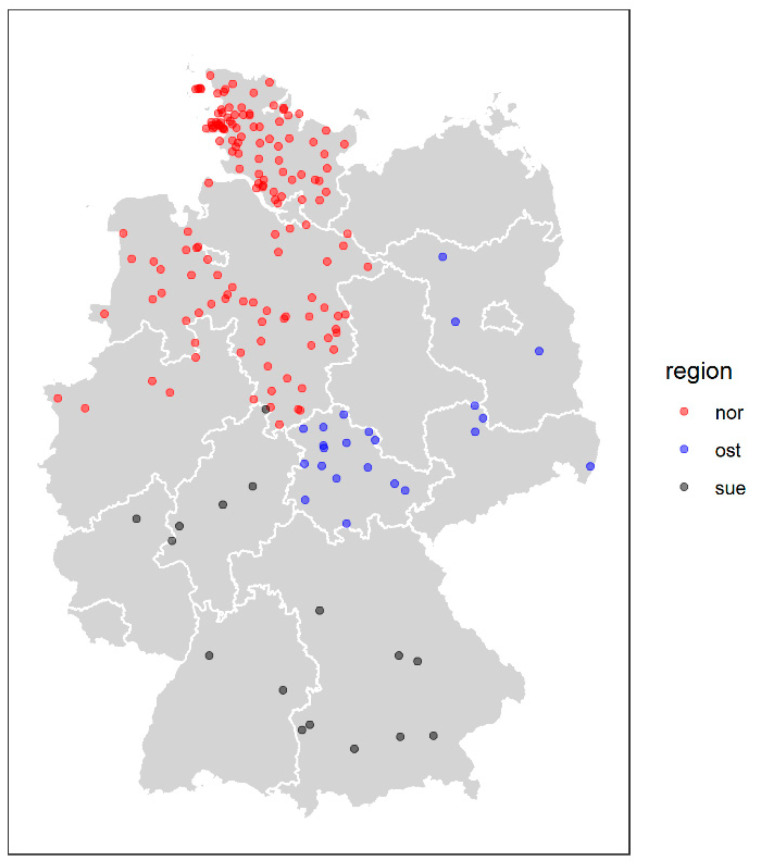
Geographical location of the 207 sheep farms sampled in Germany. Red: North Germany; blue: East Germany; black: South Germany.

**Table 1 animals-11-01102-t001:** Number of farms, mean flock size and number of swab samples collected in the three study areas.

Study Area	Number of Farms	Mean Flock Size	Number of Samples
*North Germany*	164	186	4238
*East Germany*	25	805	3463
*South Germany*	18	322	1542

**Table 2 animals-11-01102-t002:** Samples per flock, number of farms and number of sheep.

Samples Per Flock	Number of Flocks	Number of Sheep
≤40	102	1300
41–100	46	1817
101–200	17	1140
201–500	16	1627
>500	26	3359
Total	207	9243

**Table 3 animals-11-01102-t003:** Scoring system applied with the corresponding clinical findings [15].

Footrot Score	Clinical Signs
0	Healthy interdigital space and claw
1	Limited mild interdigital dermatitis, loss of hair, redness
2	More extensive interdigital dermatitis, foul smell
3	Severe interdigital dermatitis, Separation of hoof horn and dermis on the axial wall
4	Separation of hoof horn and dermis extends to the sole and the abaxial wall
5	Separation of hoof horn and dermis extends to the toe, potential loss of hoof horn

**Table 4 animals-11-01102-t004:** Numbers and percentages of footrot scores determined in 9243 German sheep from 207 flocks.

Footrot Score	Number of Sheep	Frequency (%)
0	4423	47.85
1	3548	38.39
2	705	7.63
3	346	3.74
4	142	1.54
5	79	0.85
Total	9243	100

**Table 5 animals-11-01102-t005:** Numbers and frequencies of sheep tested positive for virulent, benign *D. nodosus*, both virulotypes and negative for *D. nodosus.*

Detection of *D. nodosus*	Number of Sheep	Frequency (%)
Negative	5275	57.07
aprV2+	3585	38.79
aprB2+	188	2.03
aprV2+/aprB2+	195	2.11
Total	9243	100

**Table 6 animals-11-01102-t006:** Joint distribution of clinical signs of footrot and *Dichelobacter nodosus* strains.

Footrot Score	Negative		aprV2+		aprB2+		aprV2+/B2+		Total
	Number	%	Number	%	Number	%	Number	%	Number
0	3306	74.75	999	22.59	96	2.17	22	0.50	4423
1	1878	52.93	1466	41.32	81	2.28	123	3.47	3548
2	78	11.06	568	80.57	11	1.56	48	6.81	705
3	7	2.02	337	97.40	0	0	2	0.58	346
4	4	2.82	138	97.18	0	0	0	0	142
5	2	2.53	77	97.47	0	0	0	0	79
Total	5275	57.07	3585	38.79	188	2.03	195	2.11	9243

**Table 7 animals-11-01102-t007:** Joint distribution of maximum Footrot Scores and qPCR results on flock level.

Maximum Footrot Score	Negative		*aprV2+*		*aprB2+*		*aprV2+/B2+*		Total
	No.^1^ of Flocks	%	No.^1^ of Flocks	%	No.^1^ of Flocks	%	No.^1^ of Flocks	%	No.^1^ of Flocks
0	26	12.56	15	7.25	1	0.48	1	0.48	43
1	32	15.46	35	16.91	5	2.42	13	6.28	85
2	1	0.48	31	14.98	1	0.48	4	1.93	37
3	0	0	12	5.80	0	0	5	2.42	17
4	0	0	13	6.28	0	0	1	0.48	14
5	0	0	9	4.35	0	0	2	0.97	11
Total	59	28.50	115	55.56	7	3.38	26	12.56	207

^1^ No.: Number

## Data Availability

Restrictions apply to the availability of these data. Data were obtained from German sheep farms and are available from the authors at a reasonable request and with the permission of the sheep farmers.

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
