# Peer review of "Prevalence of Dichelobacter nodosus and Ovine Footrot in German Sheep Flocks"

_animals, 2021, doi:10.3390/ani11041102_

Round 1

Reviewer 1 Report

General comments:

This is a manuscript that presents data on the prevalence of Dichelobacter nodosus and ovine footrot in Germany. Footrot is an important disease of sheep and knowledge of the distribution of D. nodosus is important to control disease spread. Information about benign and virulent strains is also important for the same reason and also to better understand the disease. My major concern about this manuscript is the study design when it comes to the participating farms and the lack of statistics. The authors states that they announced the study at sheep meetings and on their websites. It does not sound that they were randomly chosen and hence the prevalence could be biased. There is a risk that those who signed up had problems with footrot and wanted help getting it diagnosed and that farms with healthy animals did not register to the same extent. Also more information how the farms represent the total sheep population in Germany is needed. And the data need to be more clearly presented. For example the prevalence of ovine footrot is not clearly stated in the abstract only D. nodosus prevalence.

Specific comments:

qRT-PCR is not the correct term for the method used here. RT is an accepted abbreviation for reverse transcriptase. You should use qPCR or real-time PCR instead and this throughout the manuscript.

Page 1, you should state the prevalence of ovine footrot (clinical diagnose) and not only D. nodosus prevalence here. And how can you find healthy feet in sheep? The results should be more clearly presented.

Page 2 line 51, Loss of the hard horn of the hoof or the hoof capsule but not the entire hoof I would think.

Page 3, Sample collection. Three regions were chosen. Why where these three regions chosen? More data is needed on the total population of sheep in Germany; how many sheep are there in total? How many farms? What the mean flock size? Where are they situated?

Figure 1, All farms seem to be in the north or middle of Germany to me. Since this is not clear, maybe you can color farms from the three regions with different colors so the reader can see where they are located since it is not obvious. Also a figure of the total sheep population to compare with (or if you can show that in the same figure would be nice). I can see that you have randomized the sampling within the flock but since footrot is a flock diagnosis this is not as important as to randomize which farms to sample. Also due to this I don´t think table 2 contributed much to the manuscript. I would appreciated more to have the flock sizes of the investigated farms and how they reflect the flock sizes of the total population of farms in Germany. How did you decide how many samples to take per farm? Did you take a certain percentage of the ewes and rams or? Better state this than the table I think.

Table 3, reference missing, also loss of hoof horn and not hoof.

Page 4, line 115. How do you close a cotton swab? And was it put in a carton box individually or together with other swabs? This is unclear. Also how they were sent to the laboratory and stored during shipping. And if any transport medium was used? If they were stored dry and frozen why did it take an hour to thaw them?

Page 4, line 123, The control strain JF5922, where is that from? Is it publically available? Did you not run any begative controls? What mastermix is “MasterMix” and from where is that bought? Did you measure all the samples prior to qPCR or how did you know you added 10-100 ng?

Page 5 line 130. How did you decide that the sensitivity was 99%? Or is this analytical sensitivity from Stäuble et al? Then reference is missing.

Tables 4-7, I would like to have the data on farm level also and not just individual sheep level since it often is a flock diagnosis.

I am missing a discussion on how 53% of the score 1 samples are negative for D. nodosus but still considered as footrot? Also why the majority of those are virulent and don´t cause more clinical signs? In general, why do you almost exclusively have virulent strains and so little benign strains if the majority of your lesions are less than score 3?

Page 6 line, 173. Even though 9243 samples is an impressive number the statement made that it should reflect the prevalence does not need to be accurate per se. It depends on the study design. It´s better to take fewer samples from more flocks than more samples from not so many flocks I would say since it is a flock diagnosis. No statistics seems to have been done in this manuscript what so ever not even a sample size calculation!

Page 7 line 202, The Swedish study did not examine D. nodosus prevalence merely clinical footrot where the prevalence was 5.8% on individual level. This section needs to be rewritten.

Table S2, not the correct use of detection methods (qRT PCR for example). It is not correct to draw the conclusion that the prevalence of D. nodosus in the Swedish study was 5.8%.

Author Response

  1. Reviewer

Open Review

English language and style

( ) Extensive editing of English language and style required
( ) Moderate English changes required
(x) English language and style are fine/minor spell check required
( ) I don't feel qualified to judge about the English language and style

Yes

Can be improved

Must be improved

Not applicable

Does the introduction provide sufficient background and include all relevant references?

(x)

( )

( )

( )

Is the research design appropriate?

( )

(x)

( )

( )

Are the methods adequately described?

( )

(x)

( )

( )

Are the results clearly presented?

( )

( )

(x)

( )

Are the conclusions supported by the results?

( )

( )

(x)

( )

Comments and Suggestions for Authors

General comments:

This is a manuscript that presents data on the prevalence of Dichelobacter nodosus and ovine footrot in Germany. Footrot is an important disease of sheep and knowledge of the distribution of D. nodosus is important to control disease spread. Information about benign and virulent strains is also important for the same reason and also to better understand the disease. My major concern about this manuscript is the study design when it comes to the participating farms and the lack of statistics. The authors states that they announced the study at sheep meetings and on their websites. It does not sound that they were randomly chosen and hence the prevalence could be biased. There is a risk that those who signed up had problems with footrot and wanted help getting it diagnosed and that farms with healthy animals did not register to the same extent. Also more information how the farms represent the total sheep population in Germany is needed. And the data need to be more clearly presented. For example the prevalence of ovine footrot is not clearly stated in the abstract only D. nodosus prevalence.

Specific comments:

qRT-PCR is not the correct term for the method used here. RT is an accepted abbreviation for reverse transcriptase. You should use qPCR or real-time PCR instead and this throughout the manuscript.

Amended: changed to real-time PCR throughout the manuscript

Page 1, you should state the prevalence of ovine footrot (clinical diagnose) and not only D. nodosus prevalence here. And how can you find healthy feet in sheep? The results should be more clearly presented.

Amended: Sheep with four clinically healthy feet were found through visual inspection in 47.85% of all animals included in this study. (Lines: 33-34)

  • You can find clinically healthy feet in sheep through visual inspection of all four feet of the sheep

Amended: Underrunning of horn horn on at least one foot (Scores 3-5) was detected in 567 sheep (6.13%). (Lines: 32-33)

Page 2 line 51, Loss of the hard horn of the hoof or the hoof capsule but not the entire hoof I would think.

Amended: The clinical signs can range from mild interdigital dermatitis to the complete underrunning of the hoof horn and the loss of the hoof capsule in severe cases. (Line: 55)

Page 3, Sample collection. Three regions were chosen. Why where these three regions chosen? More data is needed on the total population of sheep in Germany; how many sheep are there in total? How many farms? What the mean flock size? Where are they situated?

  • The regions were chosen, in order to structure the different geographical locations of the sheep farms in the different states of Germany.

Figure 1, All farms seem to be in the north or middle of Germany to me. Since this is not clear, maybe you can color farms from the three regions with different colors so the reader can see where they are located since it is not obvious. Also a figure of the total sheep population to compare with (or if you can show that in the same figure would be nice). I can see that you have randomized the sampling within the flock but since footrot is a flock diagnosis this is not as important as to randomize which farms to sample. Also due to this I don´t think table 2 contributed much to the manuscript. I would appreciated more to have the flock sizes of the investigated farms and how they reflect the flock sizes of the total population of farms in Germany. How did you decide how many samples to take per farm? Did you take a certain percentage of the ewes and rams or? Better state this than the table I think.

Amended: See Figure 1 (Line: 97-98)

Table 3, reference missing, also loss of hoof horn and not hoof.

Amended: Separation of hoof horn and dermis extends to the toe, potential loss of hoof horn  (Line: 109)

Page 4, line 115. How do you close a cotton swab? And was it put in a carton box individually or together with other swabs? This is unclear. Also how they were sent to the laboratory and stored during shipping. And if any transport medium was used? If they were stored dry and frozen why did it take an hour to thaw them?

  • The carton box contained 50 samples, each box is divided into 50 cases via a carton grid. Thus each swab sample is stored in the box without contact to other samples
  • The veterinarians delivered the swab samples to the laboratory
  • Thawing time was set at 1 hour to make sure the swab samples were completely thawed before further steps were done

Amended: A sterile and dry cotton swab (101 x 16.5 mm) (Sarstedt, Nümbrecht, Germany) in a tube with a screw top and without a transport medium was used to sample the interdigital skin of the foot. (Lines: 118-120)

Page 4, line 123, The control strain JF5922, where is that from? Is it publically available? Did you not run any begative controls? What mastermix is “MasterMix” and from where is that bought? Did you measure all the samples prior to qPCR or how did you know you added 10-100 ng?

Amended: The benign control strain was isolated at the Institute of Veterinary Bacteriology, University of Bern, Switzerland (Kuhnert, 2019, doi:10.17236/sat00215) (Lines: 133-135)

  • It is not publically available.

Amended: Two nontemplate samples (pyrogen-free water) as negative controls were studied in each qPCR run. (Lines: 137-138)

Amended: The 25-µl reaction mixture containing 22.5 µl TaqMan Fast Advance MasterMix (Life Technologies GmbH, Thermo Fisher Scientific, Waltham, Massachusetts) and 2.5 µl sample DNA (10-100ng) were pipetted in duplicates into a 96-well plate.(Lines: 135-137)

Page 5 line 130. How did you decide that the sensitivity was 99%? Or is this analytical sensitivity from Stäuble et al? Then reference is missing.

Amended: The assay developed by Stäuble et al. proved to be 100% specific and 100% sensitive (Stäuble, 2014) (Lines: 204-205)

Tables 4-7, I would like to have the data on farm level also and not just individual sheep level since it often is a flock diagnosis.

Amended: Table 7: Joint distribution of maximum Footrot Scores and qPCR results on flock level (Lines: 184-185)

Maximum

Footrot

 Score

negative

aprV2+

aprB2+

aprV2+/B2+

Total

No. of flocks

%

No. of flocks

%

No. of flocks

%

No. of flocks

%

No. of flocks

0

26

12.56

15

7.25

1

0.48

1

0.48

43

1

32

15.46

35

16.91

5

2.42

13

6.28

85

2

1

0.48

31

14.98

1

0.48

4

1.93

37

3

0

0

12

5.80

0

0

5

2.42

17

4

0

0

13

6.28

0

0

1

0.48

14

5

0

0

9

4.35

0

0

2

0.97

11

Total

59

28.50

115

55.56

7

3.38

26

12.56

207

I am missing a discussion on how 53% of the score 1 samples are negative for D. nodosus but still considered as footrot? Also why the majority of those are virulent and don´t cause more clinical signs? In general, why do you almost exclusively have virulent strains and so little benign strains if the majority of your lesions are less than score 3?

Amended: In our study, 3,548 sheep showed mild inflammation of the interdigital skin (Score 1) and 1,878 (52.93%) of these sheep tested negative for D. nodosus (Table 6). The early stages of footrot cannot be distinguished from interdigital dermatitis of another cause by clinical signs alone (Gilhuus, 2013). Therefore, in these cases the diagnosis of footrot can only be made in retrospect after the detection of D. nodosus. On the other hand 1,117 sheep with clinically healthy feet (Score 0) tested positive for D. nodosus. This can be explained, because footrot is a multifactorial disease. The virulence of D. nodosus has an impact on the capacity of causing severe lesions, while recent studies suggest that the presence of the aprV2 gene in D.nodosus isolates may not be the only characteristic determining the clinical outcome of lesions (Smith, 2021). In addition, environmental conditions and the inherent susceptibility play a role in the expression of clinical footrot (Raadsma, 2013). In 17 sheep flocks (8.21%) only clinically healthy sheep were present, but swab samples tested positive for D. nodosus. An explanation may be that environmental conditions were unfavourable for the bacterium. This finding emphasizes the contribution of laboratory diagnosis using qPCR in determining the presence or absence of D. nodosus and its prevalence at times with no clinical signs of footrot. Sheep farmers could apply this knowledge in order to implement certain management practices, like a close monitoring of sheep or other preventative actions to avoid future footrot outbreaks.Because no sheep flocks were revisited and no sheep were reinspected after the sampling, the development of the clinical signs of feet could not be recorded. Mild foot lesions or clinically healthy feet at the time of sampling may have been the early stages of footrot and could have progressed to higher footrot scores later on (Moore, 2005). (Lines: 206-227)

Page 6 line, 173. Even though 9243 samples is an impressive number the statement made that it should reflect the prevalence does not need to be accurate per se. It depends on the study design. It´s better to take fewer samples from more flocks than more samples from not so many flocks I would say since it is a flock diagnosis. No statistics seems to have been done in this manuscript what so ever not even a sample size calculation!

  • The number of farms included in this study (207) is in the upper range compared to previous studies
  • Ardüser, 2020: 142 sheep farms visited
  • Prosser, 2020: 164 sheep farmers submitted interdigital swabs
  • Kraft, 2020: 30 flocks
  • Moore, 2005: 39 sheep flocks
  • Gilhuus, 2013: 124 farms
  • McPherson, 2017: 40 flocks
  • Collins, 2020: 44 flocks
  • Figure S1 shows the distribution of flock prevalences of D. nodosus. The results show, that sheep flocks with a low within-flock prevalence might be categorized as “free of D. nodosus” if only few swab samples were taken from a small number of sheep, even though D. nodosus was present on a small proportion of the flock.

Page 7 line 202, The Swedish study did not examine D. nodosus prevalence merely clinical footrot where the prevalence was 5.8% on individual level. This section needs to be rewritten.

Amended: A study that was conducted on Swedish slaughter lambs in abattoirs showed a prevalence of footrot of 5.8% and it was also found that 97% of footrot affected feet tested positive for D. nodosus by PCR (Lines: 256-258)

Table S2, not the correct use of detection methods (qRT PCR for example). It is not correct to draw the conclusion that the prevalence of D. nodosus in the Swedish study was 5.8%.

Amended: Sweden: Prevalence: Footrot in 5.8% of slaughter lambs

Reviewer 2 Report

Please find comments and suggestions in attached document.

Author Response

  1. Review

This paper describes results of a field study to determine prevalence of footrot and Dichelobacter nodosus in German sheep flocks. Existing knowledge of footrot prevalence in German sheep flocks is based upon questionnaire data which primarily relied upon farmers identifying footrot in their flocks, and was published in 2012. An updated estimate for the prevalence of footrot in Germany based on both clinical diagnosis and identification of the causative agent D. nodosus is therefore valuable.

Whilst the methodology for this study appears to be robust, more information and analysis could be provided in the results section based on the data collected, and this would add value to the paper. In my opinion revisions are required for the Discussion and Conclusions sections. The discussion needs revising to provide a more detailed account of the relevance and impact of the findings of the study, particularly in relation to future options for reducing footrot prevalence in Germany. One of the conclusions is not evidenced or discussed in the paper and therefore this either needs removing or addressing in the Discussion. Details of the suggested revisions are provided below.

Summary

Line 16 – Suggest changing the word hoof to foot.

Amended: Footrot is a highly contagious foot disease in sheep and a common cause of lameness. (Line: 16)

Line 20-21 – regarding benign and virulent strains this is not correct, virulent strains can cause inflammation of the interdigital skin (Moore et al., 2005) and benign strains can cause underrunning of the hoof horn (Gilhuus et al., 2013). The link between classification of strains as virulent and benign and clinical presentation is not clear and this statement should therefore be amended or removed.

  • In the beginning of the infection, signs of interdigital dermatitis can be observed. Benign strains of D. nodosus mostly cause mild lesions, but may also cause cause underrunning of hoof horn in single animals (Gilhuus, 2013). The infection with virulent strains of D. nodosus can induce the whole spectrum of clinical manifestation from an inflammation of the interdigital skin to severe footrot with complete separation of the hoof horn from the underlying tissue under favourable environmental conditions (Allworth 2014).

Amended: Benign D. nodosus commonly causes an inflammation of the interdigital skin whereas virulent strains can lead to severe footrot with separation of hoof horn from the underlying soft tissue as the disease progresses. (Line 20-22)

Line 23 – suggest ‘interdigital skin’ rather than ‘interdigital space’. 

Amended: Benign D. nodosus commonly causes an inflammation of the interdigital skin whereas virulent strains can lead to severe footrot with separation of hoof horn from the underlying soft tissue as the disease progresses. (Line: 20)

Line 24-25 – this paper does not present evidence that sheep trading events promote spread of disease. Unless substantial changes are made to the main paper to address this, I would remove this comment from the Summary.

Amended: Due to the high prevalence of 42.93% of D. nodosus in the German sheep population, further work is required to determine measures on how to decrease the prevalence. (Line24-25)

Abstract

Line 27 – suggest ‘sheep flocks’ rather than ‘sheep farms’.

Amended: The aim of this field study was to determine the prevalence of D. nodosus in German sheep flocks. (Line: 27)

 Line 33 – suggest removing the word ‘were’ from ‘healthy feet were tested positive’.

Amended: In total, 1,117 swabs from sheep with four clinically healthy feet tested positive for D. nodosus.  (Line: 35)

Introduction

I would suggest including a brief summary of the current knowledge of footrot prevalence in Germany to evidence the need for the current study.

Major revisions

Line 47 – the statement that eradication of D. nodosus from sheep flocks is feasible is misleading. The feasibility of eradication of D. nodosus depends on the climate, production system and biosecurity status of the flock (Clifton & Green, 2016) and potentially the virulence of D. nodosus strains present in the flock (Allworth & Egerton, 2 2018). I would suggest that more discussion on this point is warranted (please also see similar comments for the Discussion).

Amended: Additional factors like the local climate conditions, production systems and biosecurity status of the flock also play a role in the elimination of D. nodosus (Green, 2008). (Lines: 51-53)

Minor revisions

Line 40 – suggest the word feet not hoof.

Amended: Ovine footrot is a highly contagious disease that affects the feet of sheep and other ruminants and it is a major challenge for sheep industries worldwide. (Line: 43)

Line 41-42 -suggest ‘aetiological agent’ rather than ‘aetiological cause’.

Amended: The aetiological agent is the gram-negative, anaerobic but aerotolerant bacterium Dichelobacter nodosus (D. nodosus) (Line: 44-45)

Line 45 – please clarify what is meant by fewer lambs, how is this different to lambing percentage? For example does this refer to fewer lambs successfully weaned? Or sold?

Amended: The disease has great economic impact on production due to decreased wool quality, reduced live-weight gain and decreased lambing percentages (Line: 47-48)

Line 56 – it could be added that warm and moist conditions also favour the transmission of D. nodosus between sheep (Graham & Egerton, 1968).

Amended: Warm and moist environmental conditions favor the transmission of D. nodosus between sheep (Graham, Egerton, 1968) and the development of clinical signs in the presence of D. nodosus (Collins, 2020). (Lines: 59-61)

Line 66-68 – this seems more like methods, it either needs to be linked to the aims of the study or moved to the methods section.

Amended: The objectives of this nationwide study were to determine the prevalence of footrot and its causative agent D. nodosus in German sheep flocks with the use of real-time PCR. In addition to the presence or absence of D. nodosus the clinical diagnosis depends on environmental factors and host susceptibility (Zanolari, 2021). Therefore, laboratory diagnosis with the use of the highly sensitive and specific real-time PCR method for determining virulent and benign D. nodosus is ideally involved (Zanolari). A broad range of different farms was included into this field study in order to reflect the diverse picture of sheep farms in Germany. (Lines: 68-75)

Methods

Major revisions

Line 94 – More detail is required on how sheep were selected for sampling. What were the minimum and maximum number of samples per flock? Please provide more detail regarding randomisation and how the number of sheep sampled per flock was decided, for example was a certain percentage of the flock sampled? Please clarify which age and gender of sheep were included in the study, for example were rams, ewes and lambs sampled in all flocks, some flocks, or were only ewes sampled? Please also clarify whether there were any flocks where all sheep were sampled and examined.

  • Ardüser, 2020: “Within each selected farm, five animals were randomly sampled. If farms kept five animals or less, all animals were sampled.” (Ardüser, 2020)

Amended: Mostly ewes and yearlings and all rams of the flock were sampled. All sheep were sampled in 60 flocks. (Lines: 104-105)

Minor revisions

 Line 92 – suggest change to past tense: ‘flock size was defined as’.

Amended: The flock size was defined as the number of ewes and rams at the time of sampling. (Line: 101)

Line 105-108 – this information does not appear to be used anywhere in the analysis therefore please consider whether it is relevant. Also, please see relevant comments on the Results section.

Amended: The electronic ear tag of every sheep was scanned and further information was recorded. (Line: 115-116)

Line 115 – was there a reason for not recording scores for all feet? Please clarify.

  • Our objectives were to determine the prevalence of footrot and its causative agent nodosus in German sheep flocks with the use of real-time PCR. Therefore, the maximum severity of the lesions present (MFS) as implemented by Marshall et al. (1991) and Stewart et al. (1982, 1983, 1985, 1986) was the most appropriate for our study. Our aim was to determine if any lesions where present on a sampled sheep in combination with the presence/absence of D. nodosus.
  • TWFS and similar methods is needed, if effects of footrot on the host are investigated or the aim is to rank sheep with qualitatively and quantitatively different lesions accurately (Whittington, 1995).

Line 129 – suggest changing to past tense.

Amended: Samples with a Ct-value < 40 were defined as being positive and a Ct ≥ 40 was interpreted as negative. (Line: 143-144)

Line 129 – please provide details regarding how samples that were not positive in duplicate were classified (were these deemed positive or negative?).

  • samples, which were not positive in duplicates were repeated and deemed either positive or negative depending on the result

Line 130 – regarding the statement about sensitivity to detect D. nodosus of 99%, I am not sure what this means. Is this a reference to the positive control? Or a previous reported performance of the assay in which case please provide a reference.

Amended: The assay developed by Stäuble et al. proved to be 100% specific and 100% sensitive (Stäuble et al., 2014) (Line: 205)

Results

Overall there is little reference made to flock size in the results, it would be useful to identify any associations between flock size and detection of footrot or D. nodosus. There is also no data provided on whether results differed between the regions of the study. Given the range of data recorded (as detailed in the methods) there is scope to provide 3 some simple summary statistics or statistical analysis that would provide a better understanding of the variation in prevalence of footrot and D. nodosus between flocks.

Major revisions

Line 148-149 – this is an interesting result, it would benefit from some additional information regarding the footrot status and size of these flocks. What was the prevalence of lesions in the flocks where D. nodosus was not detected? Were they very small flocks? If flocks were very small were all sheep sampled and examined? As a minimum please state the proportion of D. nodosus negative flocks that were considered to be free from footrot, and whether this status was based on farmer reporting or examination of all sheep by veterinarians in this study.

  • In total, in 59 sheep flocks D. nodosus was not detected.
  • See Figure S1 about the combination of flock size and flock prevalence of D. nodosus
  • Sample/data collection and examination of sheep was exclusively performed by veterinarians. (Line: 111)

Amended: Table 7: Joint distribution of maximum Footrot Scores and qPCR results on flock level (Lines: 184-185)

Maximum

Footrot

 Score

negative

aprV2+

aprB2+

aprV2+/B2+

Total

No. of flocks

%

No. of flocks

%

No. of flocks

%

No. of flocks

%

No. of flocks

0

26

12.56

15

7.25

1

0.48

1

0.48

43

1

32

15.46

35

16.91

5

2.42

13

6.28

85

2

1

0.48

31

14.98

1

0.48

4

1.93

37

3

0

0

12

5.80

0

0

5

2.42

17

4

0

0

13

6.28

0

0

1

0.48

14

5

0

0

9

4.35

0

0

2

0.97

11

Total

59

28.50

115

55.56

7

3.38

26

12.56

207

Minor revisions

Line 133 – suggest ‘from 207 sheep farms’ rather than ‘in’.

Amended: In total, 9,243 interdigital swab samples were collected from 207 sheep farms.  (Line: 147)

Line 134 – suggest change to interdigital skin rather than space.

Amended: The examination of feet showed healthy interdigital skin and claws (Score 0) in 4,423 sheep (47.85 %). (Line: 147-149)

Line 141 - I am not sure that Table 5 is really necessary, it does not add more than is provided in the text. I would suggest removing it.

Amended: Table 5 and 6 were combined à now: Table 5: Numbers and frequencies of sheep tested positive for virulent, benign D. nodosus, both virulotypes and negative for D. nodosus (Lines: 160-161)

Detection of D. nodosus

Number of sheep

Frequency (%)

Negative

5,275

57.07

aprV2+

3,585

38.79

aprB2+

188

2.03

aprV2+/aprB2+

195

2.11

Total

9,243

100

Discussion

Overall in this section, rather than simply comparing results to those from other countries, it would be useful to incorporate discussion of the control programs/strategies used in those countries and whether they are likely to be successful in Germany given the results of this study. This is definitely required if comments about control programs are to be included in the Conclusions.

Major revisions

Line 176-178 – In their recent paper McPherson et al. (2017) provided evidence that the qPCR assay developed by Stäuble et al. (2014) does not perform as well as the assay developed by Frosth et al. (2015) therefore further justification for the choice of assay or discussion of this point is required here.

  • There is no reason to change the assay developed by Stäuble et al. 2014 based on a single lab study. The assay developed by Stäuble et al. 2014 performs well and was successfully used in several international studies including effective elimination of D. nodosus in footrot affected sheep flocks. Moreover, it is the assay used in the national Swiss footrot control program.

References:

  • Greber, 2016
  • Kraft, 2020
  • Locher, 2018
  • Greber, 2018
  • Best, 2018

Line 214-222 – It might be worth discussing here that although prevalence of ID and SFR were higher in the current study than in Winter et al. (2015), almost all flocks in England are affected by footrot whereas D. nodosus was not detected in over 25% of flocks in this study. As mentioned above for Results, information on how many of these flocks were considered to free from footrot would be relevant here.

Amended: However, in this study in 59 sheep flocks (28.5%) D. nodosus was not detected, whereas in England almost all flocks were affected by footrot. Comparing these two studies, the different methods (questionnaire versus PCR) need to be considered. The prevalence of ID might be underestimated by sheep farmers in England as a mild dermatitis may be overlooked if affected sheep don’t show signs of lameness. (Lines: 278-281)

Minor revisions

Line 173 – given the previous statements about lack of randomisation and uneven distribution of sheep flocks, the statement that results reflect frequency of footrot and prevalence of D. nodosus needs some form of justification. How can the authors be certain of this? I would suggest stating that the large number of flocks and sheep studied mean that the results are likely to be robust.

Amended: Nevertheless, the combination of 9,243 analyzed swab samples and the large number of flocks suggests that the results of the study are likely to be robust. (Lines: 201-202)

Line 179-181 – This sentence fits with later discussion of D. nodosus prevalence, line 189 onwards therefore I would suggest moving it to that section. Here a statement about the prevalence of footrot would be more relevant to fit with the subsequent comments in line 182-188.

Amended: Footrot was observed in 66% of the German sheep flocks, which participated in the questionnaire, while we found that only 42 of the 207 flocks (20.29%) kept sheep with footrot scores of 3-5 at the time of sampling, which can be interpreted as being equivalent to clinical footrot. (Lines: 232-235)

Line 187 - It is not clear that the statement about sensitivity of qRT-PCR for diagnosing footrot refers to diagnosing the presence of footrot in a flock rather than in an individual foot. Suggest clarifying.

Amended: The results of the questionnaire may be an underestimation because the sheep farmers diagnosed the disease themselves in 89%, which is not as specific as the implementation of real-time PCR as a diagnostic tool to determine the presence of D. nodosus in early stages and mild cases of footrot and thus securing the diagnosis. (Lines: 236-239)

Line 196 – clarification is needed here on the difference between randomisation methods as it is not clear what is meant. See also my comment above about including more detail on randomisation in the Methods section.

  • Ardüser, 2020: “A two-stage cluster sampling strategy was conducted. For all four species, farms were randomly selected and stratified according to population size of 25 of the 26 cantons of Switzerland, one canton (Basel-Stadt, BS) having no farm registered. “

Amended: The overall lower prevalence of D. nodosus in sheep in Switzerland may be due to the different method of complete randomization using a two-stage cluster sampling strategy and only sampling five animals per farm in the Swiss study. (Lines: 251-253)

Conclusions

Line 240-242 – as discussed above regarding sensitivity of qRT-PCR and clinical diagnosis, I would suggest rephrasing to make it clear that this comparison is made at a flock level.

Amended: “Our data confirmed that D. nodosus was much more common in sheep flocks than clinical footrot and demonstrated that the PCR method in detecting D. nodosus is much more sensitive than the visual inspection of feet alone.” (Lines: 302-304)

Line 243-246 – the conclusions regarding sheep trading and events cannot be made based on the results presented here. The authors would at least need to present some discussion of studies where increased spread of footrot via these means has been demonstrated to include this in the conclusions. Currently this is the first mention of sheep trading and events in the paper.  

Amended: Due to the high prevalence of D. nodosus in the German sheep population as shown in this study, further research, including effective ways to reduce the prevalence, is needed. (Lines: 310-312)

Line 246 – 247 – if this is to be included as a conclusion from the study there needs to be greater discussion of control programs in other countries, and which programs might be most relevant for use in Germany given the observed prevalence of footrot and D. nodosus.

Amended: Due to the high prevalence of D. nodosus in the German sheep population as shown in this study, further research, including effective ways to reduce the prevalence, is needed. (Lines: 310-312)

Reviewer 3 Report

General remarks

  • The manuscript describes a field study which aimed to determine the prevalence of ovine footrot and the organism Dichelobacter nodosus in German sheep flocks. A total of 9,000 sheep from 207 flocks were inspected. Feet were scored using a previously devised scoring system. Swabs were collected from the interdigital skin of the same sheep. A qPCR assay targeting the aprV2/B2 protease genes was used to identify and differentiate virulent and benign strains of nodosus on. The organism was detected on the feet of 42.93% of sheep sampled. Of these, 90.35% were positive for virulent (aprV2) strains of D. nodosus, 4.74% were positive for benign (aprB2) strains of D. nodosus, and X% were positive for both aprV2 and aprB2 strains. The authors should be commended for the breadth of this study, as it provides very valuable information regarding the prevalence of aprV2 and aprB2-positive strains of D. nodosus in German sheep flocks. This information will obviously be very useful for control and elimination programs.

Major comments

  • Overall, this manuscript is well-written and a valuable contribution of the field. The data will be very useful for veterinarians and producers. However, I have a few issues with the way in which the data are reported. Most of the results (e.g. number sheep positive for aprV2-positive strains etc.) are reported as a % of the total number of swabs collected. This is not very informative. Instead, the data should be reported at the flock level, e.g. the number of flocks in which aprV2-positive strains were detected only, the number of flocks in which aprB2 positive strains were detected only, the number of flocks in which nodosus was not detected etc. etc. The number of swabs collected per flock varies considerably (<40 through to >500), so I assume the data cluster by flock, e.g. were aprV2-positive strains present in the flocks you sampled more intensively? If so, then reporting a large number of aprV2-positive samples may give the false impression that these strains are highly prevalent in German flocks, when in reality they are only present in a small number of flocks. Please summarise your results at the flock level in the revised manuscript. This could be done in a Supplementary Table, which you then refer to in the body of the manuscript. Tables 4 – 6 could be combined to summarise these results overall.
  • The discussion is a point of weakness in this manuscript. You have compared your results to those of other studies conducted in other countries, which is valid, but have not commented on the local implications of your results. For example, you have detected virulent nodosus in clinically healthy flocks, but have not commented on the implications of such findings for control programs, nor how this squares with our current understanding of virulence. If an aprV2-positive strain of D. nodosus is detected in a flock with no history of footrot, and environmental conditions are suitable for disease expression, should this strain be considered virulent?

Minor comments:

  • Lines 31 – 32 – “our results showed a mean prevalence of 42.93% … in the German sheep population” – does this refer to the % of flocks overall, or the % of sheep overall, or something else? Please clarify. It would be more useful to specify the % of flocks in which nodosus was detected, as well as the % of flocks in which only virulent strains/only benign strains/virulent and benign strains were detected
  • Line 32 – were all four feet healthy, or were there lesions present on one or more feet? Please clarify
  • Lines 33 – 34 – on what % of healthy feet were aprV2-positive strains detected? This is the more interesting result. Also, you seem to be using the words ‘sheep’ and ‘feet’ interchangeably. Given all four feet of each sheep were sampled, you should refer to the number or % of sheep positive rather than feet
  • Line 34 – delete either the word ‘were’ or ‘tested’
  • Line 57 – aprV2-positive strains of nodosus should be regarded as potentially virulent in the absence of phenotypic data. There are a few recent studies from Australia that demonstrate discrepancies between the genotype and phenotype of aprV2-positive strains of D. nodosus, as well as their capacity to cause severe lesions (see McPherson et al., 2017; Collins et al., 2020; Smith et al., 2021)
  • Line 94 – please mention in the abstract that one swab was used to sample all four feet of each sheep inspected
  • Lines 115-116 – in reference to my earlier comment regarding Line 32, if the highest score was recorded per animal, then I assume that all four feet of these animals were healthy?
  • Line 119 – remove the word ‘The’ from the start of the sentence
  • Lines 133-137 – it would better to report the proportion of sheep with a maximum foot score of 0, 1, 2, 3, 4, and 5 in each flock. This could be included as a supplementary table, and summarised in a new version of Table 4. Total values aren’t very informative as severe lesions would be clustered by farm/flock, and the number of samples collected per flocks varies considerably. To simplify the table, you could report, for example, the number of flocks in which there was all score 0 (i.e. all sheep were healthy), the number of flocks with mild foot lesions (score 3 or less), and the number of flocks with severe lesions (score 4 or 5).
  • The same comments as above apply to the presence/absence of nodosus and the presence absence of aprV2/B2-positive strains. It would be better to report the results at the flock level, e.g. in what proportion of flocks were aprV2-positive strains only detected etc. etc. Tables 4, 5 and 6 could be probably combined into one table.
  • Line 152 – this is a very interesting result. The histories of these flocks should be described in the Discussion. Has clinically virulent footrot ever been observed in these flocks? What does this suggest about the true virulence of these strains? What are the implications of this finding for control programs? This would support recent Australian studies which have shown that aprV2 is not necessarily a reliable indicator of virulence.

Author Response

  1. Reviewer

Open Review

English language and style

( ) Extensive editing of English language and style required
( ) Moderate English changes required
(x) English language and style are fine/minor spell check required
( ) I don't feel qualified to judge about the English language and style

Yes

Can be improved

Must be improved

Not applicable

Does the introduction provide sufficient background and include all relevant references?

(x)

( )

( )

( )

Is the research design appropriate?

(x)

( )

( )

( )

Are the methods adequately described?

(x)

( )

( )

( )

Are the results clearly presented?

( )

( )

(x)

( )

Are the conclusions supported by the results?

( )

(x)

( )

( )

Comments and Suggestions for Authors

General remarks

  • The manuscript describes a field study which aimed to determine the prevalence of ovine footrot and the organism Dichelobacter nodosus in German sheep flocks. A total of 9,000 sheep from 207 flocks were inspected. Feet were scored using a previously devised scoring system. Swabs were collected from the interdigital skin of the same sheep. A qPCR assay targeting the aprV2/B2 protease genes was used to identify and differentiate virulent and benign strains of nodosus  The organism was detected on the feet of 42.93% of sheep sampled. Of these, 90.35% were positive for virulent (aprV2) strains of D. nodosus, 4.74% were positive for benign (aprB2) strains of D. nodosus, and X% were positive for both aprV2and aprB2 strains. The authors should be commended for the breadth of this study, as it provides very valuable information regarding the prevalence of aprV2 and aprB2-positive strains of D. nodosus in German sheep flocks. This information will obviously be very useful for control and elimination programs.

Major comments

  • Overall, this manuscript is well-written and a valuable contribution of the field. The data will be very useful for veterinarians and producers. However, I have a few issues with the way in which the data are reported. Most of the results (e.g. number sheep positive for aprV2-positive strains etc.) are reported as a % of the total number of swabs collected. This is not very informative. Instead, the data should be reported at the flock level, e.g. the number of flocks in which aprV2-positive strains were detected only, the number of flocks in which aprB2 positive strains were detected only, the number of flocks in which nodosus was not detected etc. etc. The number of swabs collected per flock varies considerably (<40 through to >500), so I assume the data cluster by flock, e.g. were aprV2-positive strains present in the flocks you sampled more intensively? If so, then reporting a large number of aprV2-positive samples may give the false impression that these strains are highly prevalent in German flocks, when in reality they are only present in a small number of flocks.
  • No, flocks with an apparent footrot problem were not sampled more intensively than other flocks. The number of samples per flock depended on the flock size

  • Please summarise your results at the flock level in the revised manuscript. This could be done in a Supplementary Table, which you then refer to in the body of the manuscript. Tables 4 – 6 could be combined to summarise these results overall.

Amended: Table 5 + 6 were combined à now Table 5: Numbers and frequencies of sheep tested positive for virulent, benign D. nodosus, both virulotypes and negative for D. nodosus

Detection of D. nodosus

Number of sheep

Frequency (%)

Negative

5,275

57.07

aprV2+

3,585

38.79

aprB2+

188

2.03

aprV2+/aprB2+

195

2.11

Total

9,243

100

  • The discussion is a point of weakness in this manuscript. You have compared your results to those of other studies conducted in other countries, which is valid, but have not commented on the local implications of your results. For example, you have detected virulent nodosus in clinically healthy flocks, but have not commented on the implications of such findings for control programs, nor how this squares with our current understanding of virulence. If an aprV2-positive strain of  nodosus is detected in a flock with no history of footrot, and environmental conditions are suitable for disease expression, should this strain be considered virulent?
  • Further research is needed in order to challenge or secure the current definition of virulence and is not the aim of this study
  • We need to believe that the presence of virulent D. nodosus which doesn’t lead to clinical signs because of either unfavourable environmental conditions (for example dry summers) or management measures taken by the sheep farmer pose a risk of potentially leading to a footrot outbreak as D. nodosus is the aetiological cause of footrot.
  • This research shows that swab samples for the detection of D. nodosus are a vital part of the diagnosis of footrot and the reduction of prevalence of D. nodosus and therefore the clinical manifestation as footrot under certain circumstances as well.

Amended: The virulence of D. nodosus has an impact on the capacity of causing severe lesions, while recent studies suggest that the presence of the aprV2 gene in D. nodosus isolates may not be the only characteristic determining the clinical outcome of lesions (Smith, 2021). In addition, environmental conditions and the inherent susceptibility of the sheep play a role in the expression of clinical footrot (Raadsma, 2013). In 17 sheep flocks (8.21%) only clinically healthy sheep were present, but swab samples tested positive for D. nodosus. An explanation may be that environmental conditions were unfavourable for the bacterium. This finding emphasizes the contribution of laboratory diagnosis using qPCR in determining the presence or absence of D. nodosus and its preva-lence at times without clinical signs of footrot on sheep. Sheep farmers could apply this knowledge in order to implement certain management practices, like a close monitoring of sheep or other preventative actions to avoid future footrot outbreaks. Because no sheep flocks were revisited and no sheep were reinspected after the sampling, the development of the clinical presentation of feet could not be recorded. Mild foot lesions or clinically healthy feet at the time of sampling may have been the early stages of footrot and could have progressed to higher footrot scores later on (Lines: 212-227)

Amended: Our data confirmed that D. nodosus was much more common in sheep flocks than clinical signs of footrot and demonstrated that the PCR method in detecting D. nodosus is much more sensitive than the visual inspection of feet alone. Therefore, a widespread implementation of real-time PCR as a diagnostic tool to determine the presence or the prevalence of D. nodosus in sheep flocks with uncertain clinical appearance on feet and low footrot scores. Thus, the potential risk for sheep to develop severe footrot can be assessed, appropriate measures can be taken and hence potential future outbreaks might be prevented. (Lines: 302-309)

Minor comments:

  • Lines 31 – 32 – “our results showed a mean prevalence of 42.93% … in the German sheep population” – does this refer to the % of flocks overall, or the % of sheep overall, or something else? Please clarify. It would be more useful to specify the % of flocks in which nodosus was detected, as well as the % of flocks in which only virulent strains/only benign strains/virulent and benign strains were detected

Amended: Our results showed a mean prevalence of 42.93% of D. nodosus in the German sheep on an animal level (Lines: 31-32)

  • Line 32 – were all four feet healthy, or were there lesions present on one or more feet? Please clarify

Amended: Sheep with four clinically healthy feet were found through visual inspection in 47.85% of all animals included in this study. (Lines: 33-34)

  • Lines 33 – 34 – on what % of healthy feet were aprV2-positive strains detected? This is the more interesting result. Also, you seem to be using the words ‘sheep’ and ‘feet’ interchangeably. Given all four feet of each sheep were sampled, you should refer to the number or % of sheep positive rather than feet

Amended In total, 1,117 swabs from sheep with four clinically healthy feet tested positive for D. nodosus.  (Lines: 34-35)

  • Line 34 – delete either the word ‘were’ or ‘tested’

Amended: In total, 1,117 swabs from sheep with four clinically healthy feet tested positive for D. nodosus. (Lines: 34-35)

  • Line 57 – aprV2-positive strains of nodosus should be regarded as potentially virulent in the absence of phenotypic data. There are a few recent studies from Australia that demonstrate discrepancies between the genotype and phenotype of aprV2-positive strains of  nodosus, as well as their capacity to cause severe lesions (see McPherson et al., 2017; Collins et al., 2020; Smith et al., 2021)

Amended: On the other hand 1,117 sheep with clinically healthy feet (Score 0) tested positive for D. nodosus. This can be explained, because footrot is a multifactorial disease. The virulence of D. nodosus has an impact on the capacity of causing severe lesions, while recent studies suggest that the presence of the aprV2 gene in D. nodosus isolates may not be the only characteristic determining the clinical outcome of lesions (Smith, 2021). In addition, environmental conditions and the inherent susceptibility play a role in the expression of clinical footrot. In 17 sheep flocks (8.21%) only clinically healthy sheep were present, but swab samples tested positive for D. nodosus. An explanation may be that environmental conditions were unfavourable for the bacterium. This finding emphasizes the contribution of laboratory diagnosis using qPCR in determining the presence or absence of D. nodosus and its prevalence at times without clinical signs of footrot on sheep. Sheep farmers could apply this knowledge in order to implement certain management practices, like a close monitoring of sheep or other preventative actions to avoid future footrot outbreaks. Because no sheep flocks were revisited and no sheep were reinspected after the sampling, the development of the clini-cal presentation of feet could not be recorded. Mild foot lesions or clinically healthy feet at the time of sampling may have been the early stages of footrot and could have progressed to higher footrot scores later on (Moore, 2005). (Lines: 210-227)

  • Line 94 – please mention in the abstract that one swab was used to sample all four feet of each sheep inspected

Amended: More than 9,000 sheep from 207 flocks were screened for footrot scores using a Footrot Scoring System from 0 to 5 and sampling each sheep using one interdigital swab for all four feet of the sheep. (Line: 28-30)

  • Lines 115-116 – in reference to my earlier comment regarding Line 32, if the highest score was recorded per animal, then I assume that all four feet of these animals were healthy?
    • yes
  • Line 119 – remove the word ‘The’ from the start of the sentence

Amended: DNA was isolated with the Qiagen DNeasy Blood and Tissue Kit (Qiagen, Hilden, Germany).  (Line: 129-130)

  • Lines 133-137 – it would better to report the proportion of sheep with a maximum foot score of 0, 1, 2, 3, 4, and 5 in each flock. This could be included as a supplementary table, and summarised in a new version of Table 4. Total values aren’t very informative as severe lesions would be clustered by farm/flock, and the number of samples collected per flocks varies considerably. To simplify the table, you could report, for example, the number of flocks in which there was all score 0 (i.e. all sheep were healthy), the number of flocks with mild foot lesions (score 3 or less), and the number of flocks with severe lesions (score 4 or 5).

Amended: Table 7: Joint distribution of maximum Footrot Scores and qPCR results on flock level (Lines: 184-185)

Maximum

Footrot

 Score

negative

aprV2+

aprB2+

aprV2+/B2+

Total

No. of flocks

%

No. of flocks

%

No. of flocks

%

No. of flocks

%

No. of flocks

0

26

12.56

15

7.25

1

0.48

1

0.48

43

1

32

15.46

35

16.91

5

2.42

13

6.28

85

2

1

0.48

31

14.98

1

0.48

4

1.93

37

3

0

0

12

5.80

0

0

5

2.42

17

4

0

0

13

6.28

0

0

1

0.48

14

5

0

0

9

4.35

0

0

2

0.97

11

Total

59

28.50

115

55.56

7

3.38

26

12.56

207

  • The same comments as above apply to the presence/absence of nodosus and the presence absence of aprV2/B2-positive strains. It would be better to report the results at the flock level, e.g. in what proportion of flocks were aprV2-positive strains only detected etc. etc. Tables 4, 5 and 6 could be probably combined into one table.

Amended: Table 5 + 6  were combined à now Table 5: Numbers and frequencies of sheep tested positive for virulent, benign D. nodosus, both virulotypes and negative for D. nodosus

Detection of D. nodosus

Number of sheep

Frequency (%)

Negative

5,275

57.07

aprV2+

3,585

38.79

aprB2+

188

2.03

aprV2+/aprB2+

195

2.11

Total

9,243

100

Amended: See Table 7: Joint distribution of maximum Footrot Scores and qPCR results on flock level  (Lines: 184-185)

  • Line 152 – this is a very interesting result. The histories of these flocks should be described in the Discussion. Has clinically virulent footrot ever been observed in these flocks? What does this suggest about the true virulence of these strains? What are the implications of this finding for control programs? This would support recent Australian studies which have shown that aprV2 is not necessarily a reliable indicator of virulence.

Amended: On the other hand 1,117 sheep with clinically healthy feet (Score 0) tested positive for D. nodosus. This can be explained, because footrot is a multifactorial disease. The virulence of D. nodosus has an impact on the capacity of causing severe lesions, while recent studies suggest that the presence of the aprV2 gene in D. nodosus isolates may not be the only characteristic determining the clinical outcome of lesions (Smith, 2021). In addition, environmental conditions and the inherent susceptibility of the sheep play a role in the expression of clinical footrot (Raadsma, 2013). In 17 sheep flocks (8.21%) only clinically healthy sheep were present, but swab samples tested positive for D. nodosus. An explanation may be that environmental conditions were unfavourable for the bacterium. This finding emphasizes the contribution of laboratory diagnosis using qPCR in determining the presence or absence of D. nodosus and its prevalence at times without clinical signs of footrot on sheep. Sheep farmers could apply this knowledge in order to implement certain management practices, like a close monitoring of sheep or other preventative actions to avoid future footrot outbreaks. Because no sheep flocks were revisited and no sheep were reinspected after the sampling, the development of the clinical presentation of feet could not be recorded. Mild foot lesions or clinically healthy feet at the time of sampling may have been the early stages of footrot and could have progressed to higher footrot scores later on (Moore, 2005). (Lines: 210-227)

Round 2

Reviewer 2 Report

No further comments.

Reviewer 3 Report

I am satisfied with the authors' response to my comments